# Well-Being Leadership Training to Reduce Clinician Burnout in a Metropolitan Community Health System

**DOI:** 10.3390/healthcare13233177

**Published:** 2025-12-04

**Authors:** Tricia T. James, Alice C. Nayak, Anne M. Houff, Phani C. Kantamneni, Hsin-Fang Li, James M. Scanlan, Laura L. M. W. Chun

**Affiliations:** 1Providence Medical Group, Department of Clinician Experience, Portland, OR 97213, USA; alice.nayak@providence.org (A.C.N.);; 2Department of Palliative Care, Providence Portland Medical Center, Portland, OR 97213, USA; anne.houff@providence.org; 3Department of Critical Care, Kadlec Regional Medical Center, Richland, WA 99352, USA; phani.kantamneni@kadlec.org; 4Center for Cardiovascular Analytics, Research + Data Science, Providence Research Network, Portland, OR 97213, USA; hsinfang.li@providence.org; 5Providence Health Research Accelerator (HRA), Seattle, WA 98057, USA

**Keywords:** burnout, well-being, leadership training, emotional exhaustion, turnover intent

## Abstract

**Background:** Healthcare burnout is pervasive, necessitating more efforts to reduce it. **Objective:** To evaluate the effectiveness of well-being leadership training in reducing healthcare burnout. **Design:** The Clinician Wellness Council (CWC) leadership training consisted of 15 months of educational and small group sessions (September 2023–November 2024) with pre–post-training burnout survey comparison. **Setting:** Primary and specialty departments across a Pacific Northwest community-based hospital system. Participants comprised 22 clinicians from primary and specialty departments. Participants identified an intervention group where they would focus their leadership efforts. Those groups contained 549 clinicians, and 5439 non-intervention clinicians were controls. Intervention: Well-being leadership training. **Measures:** The Maslach Burnout Inventory (MBI) and the turnover intent questions before and after training. **Results:** Of the 22 CWC participants, 15 (68%) completed the surveys before and after training. Burnout reduction was seen (47% to 13%; *p* = 0.0253), primarily driven by decreased emotional exhaustion (EE). Among 549 intervention group clinicians, 173 completed both surveys compared to 359 of 5439 clinicians in the control group. Intervention practitioners (N = 173) showed reductions in EE items (e.g., feeling burnout, working too hard, working with people is stressful) and turnover intent. Multivariable regression analyses showed that EE reductions were associated with co-workers’ intent to leave. **Limitations:** We obtained pre- and post-training MBI measures on a subset of the intervention group co-workers rather than a majority. **Conclusions:** Participation in a well-being leadership training program consisting of education, coaching, and community building reduced burnout, increased confidence to improve their workplace, and their leadership decreased co-worker EE and turnover intent. This training creates a blueprint for reducing burnout in clinician leaders and co-workers. Primary Funding Source: A PPMC foundation grant sponsored 10 local participants. Central division funding came from system and local funds. Participants received $1000/month training reimbursement.

## 1. Introduction

Burnout, a syndrome characterized by emotional exhaustion, depersonalization, and a reduced sense of personal accomplishment [1], is associated with poor outcomes for physicians, the patients they treat, healthcare organizations, and society [2,3,4,5]. Despite growing awareness of this issue, the highest levels of burnout have been observed since the COVID-19 pandemic [6]. Multiple models and processes have been proposed to provide guidance for organizations attempting to influence well-being which include the Stanford Model of Profession Fulfillment examining workplace efficiency, culture, and individual factors [7]. The Mayo Clinic approach outlines seven well-established variables that contribute to burnout or engagement and breaks them down into individual, work-unit, organizational, and national scopes [8]. The Institute for Healthcare Improvement (IHI), National Academy of Medicine, and Mayo Clinic have all outlined suggested strategies for implementing these models across organizations [9,10,11]. Organizational leadership has specifically been identified as a critical contributor to burnout and job satisfaction for the clinicians that they lead [12]. This supports the findings from multiple systematic reviews and meta-analyses that organizational changes are more effective than individual support in mitigating burnout [13,14]. However, because these interventions are complex and resource-intensive, they are less commonly implemented. Furthermore, the unique drivers of individual and team burnout can limit the effectiveness of system wide changes which are ideally partnered with changes in work unit-level leadership [8,15,16].

Our community healthcare organization is large, with a presence in seven states and represents urban and rural sites as well as hospital and ambulatory services. There has not been a dedicated system or financial commitment to well-being initiatives. Much of well-being work has been performed on a volunteer basis by clinicians, leading to variable outcomes. With the increased proportion of clinicians that are employed by large organizations, many have lost significant autonomy and the belief that they can influence their workplace, creating an increasing divide with administrative leaders and movement toward organized labor.

Given our previous success with a hospitalist group in our organization [17], which demonstrated that training leadership behaviors and fostering clinician well-being are good intervention targets, we created the Clinician Wellness Council (CWC) as a grassroots effort to reduce burnout. We recruited participants to become well-being leaders, with leadership focused on influencing the well-being of their teams defined as a top priority. Our goal was to equip practicing clinicians with coaching, education and community to develop interventions to reduce group workplace stressors. We hypothesized that this would improve the workplace experience for participants, increase support for clinicians performing well-being work, and decrease work-unit burnout. We believe that this would be possible in our large complex, community hospital organization without additional well-being infrastructure or long-term financial commitment. This study is unique in that it primarily targeted practicing clinicians without formal leadership titles to take on a new role of prioritizing the well-being of the group and providing a model for engaging front-line clinicians and expanding leaders within an organization.

## 2. Methods

The study was conducted in accordance with the Declaration of Helsinki, and the protocol STUDY2023000731 was approved by the Providence Health and Systems IRB on 4 October 2023. The formal IRB review deemed that informed consent for participation was not required, and a waiver of consent was granted. All medical staff members at Providence Portland Medical Center (PPMC), including employed and affiliated groups, were invited to participate in the CWC. Interest was solicited through email communications. Specific invitations to leaders were also extended to ensure that a variety of departments were represented. To increase program impact, we invited 10 additional participants from the Providence Central Division through communication with Chief Medical Officers, which extended reach through rural areas and 2 additional states. The program was targeted at practicing clinicians without formal group leadership roles. Initially, there were 22 participants, including an existing hospitalist group wellness leader who had 0.1 FTE for her leadership role. The palliative care group had 2 participants who opted to share their experience and leadership expectations. One participant left the program after 3 months because her independent surgery group was dissolved. Of the remaining 21 participants, 2 were practicing medical directors, but the remainder were practicing clinicians in their group.

Training Program: Participation was required by attending a full-day in-person kickoff education event in addition to monthly educational sessions and small group sessions for 15 months. An identical educational session was held at 2 different time points each month to encourage live participation. Educational topics included team building, navigating conflict, understanding control, influence and concern, and other leadership tools. Small groups were decided based on common settings—outpatient, hospitalists, medical subspecialists, and hospital-based specialties—and were intended to provide space for peer support and coaching, which have been validated as effective tools to reduce burnout [18,19,20]. A complete list of groups represented, and the educational topics are in Appendix A. Participants were asked to identify their unique working group for which they were expected to recognize specific workplace stressors and develop interventions in their sphere of control.

A grant from the PPMC foundation was obtained to sponsor 10 local participants. Funding for the participants from the central division was obtained from a combination of system and local funds. Each participant was given $1000/month as training reimbursement throughout the length of the program. Participants were expected to work an average of 2 h per week, participating in the program elements in addition to leading well-being efforts for their team. Their clinical expectations were not decreased during this time. The training reimbursement acknowledged the significant time and energy commitment required for the work and that both were valued.

Surveys: Each participant was asked to fill out the same program metric survey at the kickoff event and at the final educational session to assess the effectiveness of the program’s objectives. Answers were in the form of a 5-point Likert scale ranging from ‘strongly disagree’ to ‘strongly agree’.

Surveys were sent out in October of 2023 and 2024 to all CWC participants and their identified intervention groups. All clinicians on the medical staff that were not in an intervention group at these sites were invited to fill out the same survey and served as the control group. Surveys were sent via Redcap, allowing individual responses to be tracked.

Physician burnout was evaluated using the Maslach Burnout Inventory (MBI), which involved 3 domains: emotional exhaustion (EE) (9 items), depersonalization (DP) (5 items), and professional accomplishment (PA) (8 items). Burnout was defined as patients having “high” EE (sum of scores > 27) or “high” DP (sum of scores > 10). Job satisfaction was evaluated on a scale of 0–100. Four turnover intent questions were asked as well as perceptions about well-being support at the local and system level and a comment box asking them to share their largest drivers of burnout.

Intentions to stay in their current job was assessed via four questions: “How often have you considered leaving your job?”, “How often are you frustrated when not given the opportunity at work to achieve your personal work-related goals?”, “How often do you dream about getting another job that will better suite your personal needs?”, and “How likely are you to accept another job at the same compensation level should it be offered to you?”. They were evaluated using a 5-point Likert scale, with a higher score representing higher likelihood frequencies.

Statistics: Clinician characteristics and change in turnover intent questions were compared between control and intervention groups. Survey scores were summarized using means and standard deviations (SD) and compared using two-sample *t*-tests. Categorical variables were summarized using frequencies and percentages and compared using Pearson’s chi-square tests.

To compare clinicians’ pre- and post-intervention responses, clinicians with complete responses were included and compared using paired *t*-tests. To determine the impact of intervention on MBI scores over time, difference-in-differences (DiD) analyses were performed. In the DiD framework, the change in score from pre- to post-intervention was compared between the control and intervention group. The effect of the intervention was estimated using DiD estimates, which was calculated by subtracting the score change in the control group from that of the intervention group. The difference was compared using two-sample independent *t*-tests.

The relationship between changes in emotional exhaustion and turnover intentions was assessed using linear regression models. Each turnover intent question was treated as the outcome while adjusting for all measures of emotional exhaustion. To identify the most important predictors, a stepwise regression model was used with significance levels for entry (SLE) and stay (SLS) set at 0.15 and 0.1, respectively. The final regression model was adjusted for clustering by the state of the facility sites (MT, OR, and WA). Results were reported using regression coefficients and standard errors, and the model’s performance was evaluated using R^2^. This approach considered potential confounding factors while retaining the most relevant predictors, yielding an efficient and robust final mode.

In addressing multiple hypothesis testing, the Benjamini–Hochberg procedure was applied to adjust the *p*-values obtained from multiple comparisons, maintaining a false discovery rate of 10%. In the table, *p*-values that remain significant after this adjustment are marked with an asterisk (*). All analyses were conducted using SAS Enterprise Guide 8.3.

Role of Funding Source: A grant was obtained to sponsor 10 PPMC participants as mentioned above. It did not play a role in the study design, conduct, or reporting.

## 3. Results

Of the original 22 participants in the CWC, 15 completed the survey at both time points. Among the 15 participants, there was a significant reduction in burnout, decreasing from 47% to 13% (*p* = 0.03), primarily driven by a decrease in emotional exhaustion after the intervention (Table 1).

We asked the CWC participants to fill out a survey to evaluate our program objectives. A total of 21 individuals responded before the program started, and 18 responded at the program’s conclusion. There was notable improvement in 9 of 14 measures (Table 2). The percentage of participants who responded to agree or strongly agree to each objective also increased noticeably from pre- to post-program (Figure 1).

Baseline survey response rates were 54.1% for intervention groups (N = 549) and 15.4% for control groups (N = 5439). Repeat survey response rates were 44.7% for the intervention groups (N = 515) and 13.0% for the control groups (N = 5642). Of the respondents, 173 from the intervention groups and 359 from the control groups completed the survey at both time points, allowing for individual tracking.

Table 3 shows the demographic information for individuals in the intervention and control groups that filled out the survey at both time points. Table 4 shows the change in the emotional exhaustion subscale of the MBI for individuals in the intervention and control groups over time. Members of the intervention group demonstrated noticeable decrease in emotional exhaustion in three specific questions: “I feel burned out from my work” (DiD = −0.27), “I feel I’m working too hard on my job” (DiD = −0.60), and “Working with people directly puts too much stress on me” (DiD = −0.27), while “I feel I’m working too hard on my job” remained statistically significant after applying the Benjamini–Hochberg procedure. There was no statistically significant difference in the depersonalization or personal accomplishment items of the MBI for the same individuals. In addition, the intervention group showed improvement in three of four turnover intent questions compared to the control (Table 5).

We examined if the change in emotional exhaustion questions in the MBI was associated with the change in turnover intent using stepwise linear regression models. We found that the questions “Working with people all day is really a strain for me”, “I feel burned out from my work”, “I feel frustrated by my job”, “I feel I’m working too hard on my job”, “working with people directly puts too much stress on me”, and “I feel like I’m at the end of my rope” were positively associated with turnover intent questions as outlined in Table 6.

## 4. Discussion

Participants in a well-being leadership program showed a remarkable decrease in clinician burnout. This was despite a significant increase in their workload as the program expectations were estimated to be an average of 2 h per week of participation and intervention development and implementation and with no change in their clinical duties. There was a significant improvement in the emotional exhaustion questions regarding “feeling used up”, fatigued, frustrated, and that they are working too hard at their job (Table 1). Our post-program evaluation showed that participants gained considerable confidence in their ability to positively influence their workplace (Figure 1). Significant changes were observed in “the understanding of how burnout is measured” (52% to 100%), “recognizing burnout in self and others (67% to 100%), knowing what is in the sphere of control and influence at work (60% to 100%), the ability to design well-being interventions (35% to 100%), and feeling like my team well-being is improving (30% to 72%). We found it encouraging that the single greatest intervention change was noted in the ability to design well-being interventions. This speaks to the importance of clinicians feeling effective in their workplace and being able to create positive change there. A total of 18 of the 21 clinicians in the program were employed by the organization, and the remaining 3 were part of independent groups that were affiliated.

Our program helped create a model on how to empower workers in a large system to build autonomy at the work-unit level. Part of our education focused on effective advocacy and how to work productively with leaders in a large and complex organization. Overall, we saw a noteworthy mindset shift from disempowerment and blaming to embracing a yes/and mindset. The yes/and mindset comes from improvisational theater in which “yes/and” is viewed as productive framework for collaboration and brainstorming, as opposed to “yes/but” which often stops the conversation. The “yes/and” framework helps clinicians view system challenges as potential opportunities rather than obstacles.

There were significant differences in survey response rates between the intervention and control groups at both time points. The intervention groups had a well-being leader directly encouraging them to fill out the surveys to help inform tangible workplace changes, whereas the control groups did not have this communication, which likely explains the discrepancy. Recent data shows that survey nonresponders are more likely to experience burnout and leave the organization and, therefore, our control group data is likely underestimating true prevalence [21,22]. Although we did not see a significant decrease in overall burnout rates for the intervention groups that were the recipients of the well-being interventions, we did see a significant decrease in several elements of emotional exhaustion compared to the control group (Table 4). While the gold standard for burnout is often calculated by combining the three subscales of the Maslach Burnout Inventory, there has also been relevant work on the three separate subscales (emotional exhaustion, depersonalization, and personal accomplishment).

Across a variety of populations, relevant studies, to date, strongly suggest that EE is very related to a number of important medical outcomes and may be the most important of the MBI subscales. Emotional exhaustion in clinicians has been connected to decreased patient satisfaction, infection rates, standardized mortality ratios, and disruptive behavior [23,24,25,26,27,28,29]. Tawfik reported that recent medical errors are directly proportional to burnout levels. Emotional exhaustion was the most important, with high emotional exhaustion associated with 18% of recent medical errors and low emotional exhaustion associated with 4.6% of recent errors. Even within work units with very good to excellent safety grades, doctors with burnout were three times more likely to make medical mistakes [30]. Emotional exhaustion was found to mediate intent to leave in hospital nurses [31]; and supervisor support decreased intent to leave by reducing EE in mental healthcare providers [32]. Subscale comparison of the Maslach Burnout Inventory in emergency nurses showed that EE was the most predictive of intent to leave, and twice as predictive as a sense of accomplishment [33]. In Japanese perioperative nurses, EE predicted both job satisfaction and intent to leave [34]. Focusing on physician burnout, 43 papers suggested that EE was the most related to negative outcomes [35]. In the context of these previous results, we find it encouraging that our most consistent outcomes were changes in EE items for both the participant leaders and the intervention groups.

Consistent with this literature review, our results showed that our training affected five EE items (four significantly, one borderline), while two items from personal accomplishment showed borderline changes, and none of the depersonalization items showed any changes. In the context of our findings and the previous literature, we believe the goal of interventions like ours should be to reduce mean emotional exhaustion, not simply to eradicate all measures of burnout. Our hypothesis is that overall group burnout changes will take longer as new leaders were still learning new skills and strategies throughout the program that had not yet been fully implemented. Each intervention group now has an embedded well-being team leader that can easily interpret their well-being data, understand the context, and develop the next steps. Furthermore, many of the interventions developed are laying the foundations of gradual change such as shifting culture and building community. The impact on the intervention groups was less than our previous pilot program with one hospitalist group which started in July of 2020 [17]. There are likely several reasons for this, including a critical need for interventions during the COVID-19 pandemic, funding for FTE, and partners within the hospitalist group that enabled more dedicated time to develop many group-specific interventions that were not possible with the CWC. We also suspect that there may be certain “lagging” effects of our leadership training that may not manifest for several years. For instance, physicians who express “intent to leave” are not typically quitting their jobs 1–2 months after expressing that sentiment but are much more likely to leave in the next two years.

We found significant relationships between multiple emotional exhaustion items from the MBI and four validated turnover intent questions. These findings are consistent with the previous literature showing the cost savings of decreasing emotional exhaustion and, therefore, workplace turnover [24,25] (Table 6). Our intervention groups showed improvement in the emotional exhaustion items most closely linked to turnover intent. Furthermore, in our intervention groups, we observed a significant improvement in the question “How often do you dream about getting another job that will better suite your personal needs?” with a large improvement in two of the other three turnover intent questions (Table 5).

Reducing turnover intent is critical for all hospital systems due to the substantial costs associated with replacing and recruiting physicians. Estimates of Stanford recruitment costs in 2013 ranged between 268,000 and 958,000, depending on specialty experience and expertise [36]. Using panel estimates of the number of Medicaid and non-Medicaid patients, Sinsky estimates $86,336 additional patient costs in the first year after the physician leaves their practice, independent of the costs to recruit and replace a physician [37]. As the physician shortage is anticipated to worsen, prioritizing the clinician experience and maximizing retention will be essential for organizational success. Many organizations may not be able to fill positions after individuals leave.

Empowering clinicians to become effective change agents and develop grassroots solutions for workplace stressors will ultimately be cost-saving by decreasing turnover and increasing revenue production. It will also encourage increased local autonomy and effective partnership with executive leaders, likely decreasing unionization impetus and creating a more collaborative atmosphere. Clinicians are key stakeholders in healthcare, and their perspective and engagement are critical for developing a sustainable model moving forward. Next steps in our organization include expanding this well-being leadership training to additional teams, creating opportunities for our existing leaders to learn and implement the same skills and strategies, and considering how we can bring these concepts to every team and clinician.

There are several limitations to this study. Our intervention was limited to one large community health system, and while we hypothesize the approach of well-being leadership education, coaching, and community building is transferable, this needs to be verified in different settings and will likely need to be adjusted for each unique organizational structure and culture. Additionally, we solicited volunteers for participation, selecting individuals and groups that were more open to engaging in process improvement. They also received a $1000/month training reimbursement in acknowledgement of the significant time demands, but this has the potential to introduce bias in financial motivation for joining. Further studies are needed to elucidate best practices for groups that are not open to change and how to best compensate clinicians for their time spent in leadership training activities. Finally, significant differences in the response rates between intervention and control groups were observed, emphasizing the impact of personal communication and action from a well-being leader in soliciting responses, which limits the generalizability of our control data. For the regression analysis, we included only complete responses to ensure the integrity of the data, which means our conclusions are especially applicable to the subset of participants who completed both pre- and post-intervention surveys.

## 5. Conclusions

A 15-month well-being leadership training program in a community hospital system was an inexpensive strategy that significantly decreased participant burnout and increased their confidence in their ability to effect change. The changes in participant burnout were all in EE domain items. While the intervention groups associated with the participant leaders did not show significant changes in overall burnout scores, several EE items did show significant improvement. We view this as promising, because the bulk of previous research indicates that EE is correlated with adverse outcomes.

## Figures and Tables

**Figure 1 healthcare-13-03177-f001:**
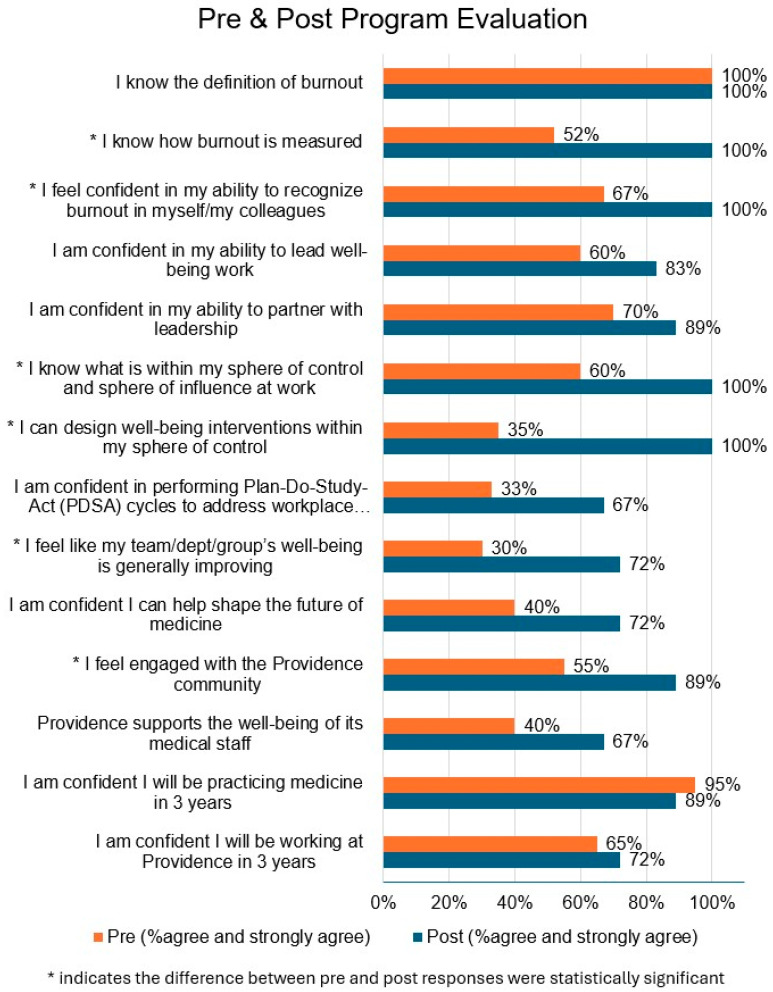
Program evaluation between pre- and post-time periods for council members.

**Table 1 healthcare-13-03177-t001:** MBI, burnout rates, and job satisfaction for council members.

Mean (SD)	Complete Pre (N = 15)	Complete Post (N = 15)	*p*-Value
**Emotional Exhaustion**			
I feel emotionally drained from my work.	2.4 (1.3)	2.3 (1.5)	0.709
I feel used up at the end of the workday.	3.3 (2)	2.4 (1.5)	0.0479
I feel fatigued when I get up in the morning and have to face another day on the job.	2.4 (1.8)	1.3 (1.3)	0.0061
Working with people all day is really a strain for me.	1.7 (1.6)	1.1 (1.3)	0.199
I feel burned out from my work.	2.3 (1.3)	1.8 (1.6)	0.1038
I feel frustrated by my job.	3.5 (1.8)	2.3 (1.7)	0.0364
I feel I’m working too hard on my job.	3.1 (2)	2.3 (1.8)	0.0406
Working with people directly puts too much stress on me.	1 (1.3)	0.4 (0.6)	0.1442
I feel like I’m at the end of my rope.	1.5 (2)	0.7 (1.7)	0.0679
**Depersonalization**			
I feel I treat some recipients as if they were impersonal objects.	0.9 (1)	0.8 (0.8)	0.547
I’ve become more callous toward people since I took this job.	1.6 (1.5)	1.7 (1.8)	0.8792
I worry that this job is hardening me emotionally.	2.1 (1.6)	1.5 (1.7)	0.1989
I don’t really care what happens to some recipients.	0.5 (1.1)	0.6 (0.7)	0.6976
I feel recipients blame me for some of their problems.	2.3 (1.4)	2.6 (2.1)	0.43
**Personal Accomplishment**			
I can easily understand how my recipients feel about things.	5.1 (1.7)	5.7 (0.6)	0.1362
I deal very effectively with the problems of my recipients.	5.2 (1.4)	5.3 (1.8)	0.876
I feel I’m positively influencing other people’s lives through my work.	4.9 (1.7)	5.5 (0.8)	0.0697
I feel very energetic.	4.2 (1.7)	4.7 (1.2)	0.1502
I can easily create a relaxed atmosphere with my recipients.	5.5 (0.8)	5.7 (0.5)	0.2722
I feel exhilarated after working closely with my recipients.	4.1 (1.7)	4.5 (1.3)	0.384
I have accomplished many worthwhile things in this job.	4.3 (1.6)	5.3 (0.8)	0.0552
In my work, I deal with emotional problems very calmly.	5.4 (1.1)	5.4 (1.1)	1
**Satisfied with job, mean (sd)**	74.1 (17.2)	76.9 (24.6)	0.6781
**Burnout**	7 (47%)	2 (13%)	0.0253

**Table 2 healthcare-13-03177-t002:** Program evaluation results for council members.

Questions	Pre (N = 21)	Post (N = 18)	Difference	*p*-Value
I know the definition of burnout	4.4 (0.5)	4.7 (0.5)	0.3 (0.5)	0.0787
I know how burnout is measured	3.5 (0.9)	4.6 (0.5)	1.1 (0.7)	<0.0001 *
I feel confident in my ability to recognize burnout in myself/my colleagues	3.7 (0.8)	4.7 (0.5)	1.0 (0.7)	0.0002 *
I am confident in my ability to lead well-being work	3.6 (0.8)	4.2 (0.7)	0.6 (0.8)	0.0295 *
I am confident in my ability to partner with leadership	3.9 (0.9)	4.3 (0.7)	0.4 (0.8)	0.1111
I know what is within my sphere of control and sphere of influence at work	3.7 (1)	4.6 (0.5)	1.0 (0.8)	0.0007 *
I can design well-being interventions within my sphere of control	3.2 (0.8)	4.3 (0.5)	1.1 (0.7)	<0.0001 *
I am confident in performing Plan-Do-Study-Act (PDSA) cycles to address workplace stressors	3 (1.2)	3.9 (0.9)	0.9 (1.1)	0.0153 *
I feel like my team/dept/group’s well-being is generally improving	2.9 (1)	3.8 (0.8)	0.9 (0.9)	0.0026 *
I am confident I can help shape the future of medicine	3.2 (1)	3.9 (1)	0.7 (1.3)	0.0383 *
I feel engaged with the Providence community	3.7 (1.1)	4.2 (0.6)	0.5 (0.9)	0.0833
Providence supports the well-being of its medical staff	3 (1.1)	3.7 (0.8)	0.7 (0.9)	0.0212 *
I am confident I will be practicing medicine in 3 years	4.5 (0.6)	4.4 (0.8)	−0.1 (0.7)	0.7984
I am confident I will be working at Providence in 3 years	3.9 (0.7)	3.9 (0.8)	0.04 (0.8)	0.8808

* *p*-values that remained significant after Benjamini–Hochberg procedure to control for the false discovery rate at 10%.

**Table 3 healthcare-13-03177-t003:** Demographic information for individuals in control and intervention groups that responded to both surveys.

	Control	Intervention	*p*-Value
N	359	173	
Age			0.0002
<30	4 (1%)	0 (0%)	
30–39	70 (20%)	53 (31%)	
40–49	143 (40%)	62 (36%)	
50–59	88 (25%)	50 (29%)	
60–70	54 (15%)	7 (4%)	
Gender			0.0366
Male	135 (39%)	81 (49%)	
Female	213 (61%)	86 (52%)	
Race			0.0297
Asian	47 (15%)	38 (25%)	
Other	10 (3%)	6 (4%)	
White	258 (82%)	110 (71%)	
Ethnicity			
% Hispanic	17 (5%)	4 (2%)	0.1824
Years since residency?			0.0796
2 or less	25 (7%)	11 (6%)	
3–5	41 (12%)	29 (17%)	
6–10	49 (14%)	29 (17%)	
11–15	77 (22%)	42 (25%)	
16–20	59 (17%)	30 (18%)	
>20	103 (29%)	30 (18%)	
Current FTE			0.0335
1	211 (59%)	121 (71%)	
0.9–0.99	31 (9%)	11 (6%)	
0.8–0.89	36 (10%)	20 (12%)	
0.7–0.79	38 (11%)	9 (5%)	
0.6–0.69	19 (5%)	3 (2%)	
0.6 or less	21 (6%)	6 (4%)	

**Table 4 healthcare-13-03177-t004:** Emotional exhaustion for intervention and control groups.

Mean (SD)	Control (N = 359)	Intervention (N = 173)	Diff in Diff	*p*-Value
	Post-Pre	Post-Pre		
**Emotional Exhaustion**				
I feel emotionally drained from my work.	−0.09 (1.38)	−0.12 (1.29)	−0.04 (0.13)	0.7671
I feel used up at the end of the workday.	−0.12 (1.41)	−0.29 (1.39)	−0.17 (0.13)	0.1791
I feel fatigued when I get up in the morning and have to face another day on the job.	−0.12 (1.5)	−0.3 (1.53)	−0.19 (0.14)	0.1894
Working with people all day is really a strain for me.	−0.04 (1.48)	−0.09 (1.56)	−0.05 (0.14)	0.7183
I feel burned out from my work.	−0.03 (1.32)	−0.3 (1.31)	−0.27 (0.12)	0.0256
I feel frustrated by my job.	−0.09 (1.39)	−0.29 (1.49)	−0.2 (0.13)	0.1418
I feel I’m working too hard on my job.	0.21 (1.71)	−0.39 (1.66)	−0.6 (0.16)	0.0002 *
Working with people directly puts too much stress on me.	0.07 (1.34)	−0.2 (1.47)	−0.27 (0.13)	0.048
I feel like I’m at the end of my rope.	0.01 (1.43)	−0.24 (1.32)	−0.25 (0.13)	0.0536

* *p*-values that remained significant after Benjamini–Hochberg procedure to control for the false discovery rate at 10%.

**Table 5 healthcare-13-03177-t005:** Change in turnover intent for intervention and control groups.

	Control	Case	*p*-Value
	359	173	
How often have you considered leaving your job? *	0.07 (1.07)	−0.09 (0.86)	0.0569
How often are you frustrated when not given the opportunity at work to achieve your personal work-related goals? *	−0.11 (1.20)	−0.09 (1.12)	0.861
How often do you dream about getting another job that will better suite your personal needs? *	0.09 (1.07)	−0.11 (1.15)	0.0495 ^*
How likely are you to accept another job at the same compensation level should it be offered to you? **	0.18 (1.25)	−0.04 (1.19)	0.0608

* 1 = Never, 5 = Always; ** 1 = highly unlikely, 5 = highly likely; ^* *p*-values that remained significant after Benjamini–Hochberg procedure to control for the false discovery rate at 10%.

**Table 6 healthcare-13-03177-t006:** Stepwise linear regression results predicting turnover intents using emotional exhaustion questions.

Stepwise Selection	Leave Job (R^2^ = 0.22)		Frustrated (R^2^ = 0.11)		Dream Other Job (R^2^ = 0.13)		Accept Another Job (R^2^ = 0.14)	
	Est (Std)	*p*-Value	Est (Std)	*p*-Value	Est (Std)	*p*-Value	Est (Std)	*p*-Value
Working with people all day is really a strain for me.							0.07 (0.02)	0.0521
I feel burned out from my work.	0.09 (0.05)	0.1962			0.12 (0.04)	0.091		
I feel frustrated by my job.	0.19 (0.05)	0.04	0.17 (0.01)	0.0013			0.06 (0.06)	0.4229
I feel I’m working too hard on my job.			0.06 (0.01)	0.018	0.18 (0.04)	0.029	0.10 (0.03)	0.0531
Working with people directly puts too much stress on me.	0.05 (0.04)	0.3076						
I feel like I’m at the end of my rope.	0.12 (0.02)	0.0221	0.10 (0.04)	0.0872			0.19 (0.03)	0.0131

## Data Availability

The data presented in this study are available on request from the corresponding author. The data are not publicly available due to privacy restrictions.

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
