# Peer review of "Well-Being Leadership Training to Reduce Clinician Burnout in a Metropolitan Community Health System"

_healthcare, 2025, doi:10.3390/healthcare13233177_

Round 1

Reviewer 1 Report

Comments and Suggestions for Authors

The subject of the manuscript is very interesting, while the study design is very effective. The paper has a good structure and is generally well written. The abstract is very informative and probably contains more information than usual. On the other hand, there are certain elements that need to be improved.
1. The Introduction section is not sufficiently developed, and contains some unclear notations and definitions.
The literature review is presented in only one paragraph with a very reduced content. For example, it states as follows "Multiple models and processes have been proposed to provide guidance for organizations attempting to influence well-being (7-13)." Seven references are sublimated in one sentence. Instead of this approach, a broader description is needed, as well as a clear connection between leadership and burnout. Also, it is necessary to define and explain "Well-Being Leadership", and then the connection between Well-Being Leadership and work-related outcomes, including burnout.
The context is not clear enough. The text states as follows "In our large, complex, community healthcare organization..." A more detailed explanation of the context is needed, especially the following statement "Indeed, our organization was a part of the largest clinician strike in US history." This part of the text contains a significant presence of a personal note, which is not suitable for academic writing and therefore needs depersonalization. It seems as if the text was written by principals.
At the end of the Introduction section, it is necessary to state the novelty of the study and the research gap.
2. Tables and graphs should be positioned in the text, instead of at the end of the text. This can contribute to the clarity and better understanding of the results obtained.
3. In the conclusion, a personal attitude appears again ("We desperately need clinicians to be a part of system change to ensure that medicine is sustainable, so our community can receive the care that they deserve.) It is necessary to carefully revise the text and depersonalize it in every part of the text where it is present. Expressing personal attitudes is not characteristic of academic writing.
4. What are the limitations of the study? At the end of the Discussion section, all real and relevant limitations of the study should be clearly stated and explained, as well as guidelines for future research.
Although the research presented in the manuscript is essentially a case study, it needs to be put into context and generate generalizations based on this, with practical value for a wider context, not only for the specified institution.

Author Response

Thank you very much for taking the time to review this manuscript. We are grateful for your thoughtful feedback.  Please find the detailed responses below and the corresponding revisions/corrections highlighted/in track changes in the re-submitted files.  

Comment 1:  

The Introduction section is not sufficiently developed and contains some unclear notations and definitions. The literature review is presented in only one paragraph with a very reduced content. For example, it states as follows "Multiple models and processes have been proposed to provide guidance for organizations attempting to influence well-being (7-13)." Seven references are sublimated in one sentence. Instead of this approach, a broader description is needed, as well as a clear connection between leadership and burnout. Also, it is necessary to define and explain "Well-Being Leadership", and then the connection between Well-Being Leadership and work-related outcomes, including burnout. The context is not clear enough. The text states as follows "In our large, complex, community healthcare organization..." A more detailed explanation of the context is needed, especially the following statement "Indeed, our organization was a part of the largest clinician strike in US history." This part of the text contains a significant presence of a personal note, which is not suitable for academic writing and therefore needs depersonalization. It seems as if the text was written by principals. At the end of the Introduction section, it is necessary to state the novelty of the study and the research gap. 

Response 1: 

Multiple modifications were made to the introduction.  The discussion of existing models and processes were expanded upon, the connection between leadership and well-being was more clearly stated and “well-being leadership” was defined.  We clarified our organization and context and removed personal language.  We added the novelty of the study and where it fills a current gap at the end of the introduction.   

Comment 2: Tables and graphs should be positioned in the text, instead of at the end of the text. This can contribute to the clarity and better understanding of the results obtained. 

Response 2: 

Tables and figures were moved into the body of the results section to help clarity and ease of understanding. 

Comment 3: 

In the conclusion, a personal attitude appears again ("We desperately need clinicians to be a part of system change to ensure that medicine is sustainable, so our community can receive the care that they deserve.) It is necessary to carefully revise the text and depersonalize it in every part of the text where it is present. Expressing personal attitudes is not characteristic of academic writing. 

Response 3: 

This sentence was re-written to make it more characteristic for academic writing. 

Comment 4: 

  1. What are the limitations of the study? At the end of the Discussion section, all real and relevant limitations of the study should be clearly stated and explained, as well as guidelines for future research.
    Although the research presented in the manuscript is essentially a case study, it needs to be put into context and generate generalizations based on this, with practical value for a wider context, not only for the specified institution.

Response 4:  

An additional paragraph was added at the end of the discussion section discussing limitations and suggestions for future research within a wider context.   

Reviewer 2 Report

Comments and Suggestions for Authors

I thank the authors and editors for the opportunity to review this magnificent work. I have just two comments:

  1. What method was used to calculate the sample?
  2.  I think it would be useful to describe in the abstract the methods used to reduce burnout. 

Author Response

Thank you very much for taking the time to review this manuscript. We are grateful for your thoughtful feedback.  Please find the detailed responses below and the corresponding revisions/corrections highlighted/in track changes in the re-submitted files.  

Comment 1: 

What method was used to calculate the sample? 

Response 1: 

No sample size calculation was done for this study. All medical staff members at PPMC were invited to participate. Interest was solicited through email.  

Comment 2: 

I think it would be useful to describe in the abstract the methods used to reduce burnout. 

Response 2: 

Added clarification of key elements of the training program that led to decreased burnout in the abstract. 

Reviewer 3 Report

Comments and Suggestions for Authors

Thank you very much for giving me the opportunity to review this manuscript.

The study addresses an important question and shows promising signals, notably on Emotional Exhaustion items. However, there are several methodological, analytic, and reporting issues that must be addressed before the manuscript is suitable for publication.

The program recruited clinicians by invitation and also provided $1000/month training reimbursement to participants. This high monetary incentive could influence responses. Need more reasoning for this bias in your limitations and if possible,  present baseline characteristics like age, gender, years in practice etc. to compare participants with non-participants and intervention with control clinicians.

Another concern is the low and differential response rates i.e. baseline response rates differ widely, because intervention group is      54.1 percent whereas control group is 15.4 percent. The n=15 paired participants is very small for inferential claims. Also, some denominators in the Methods/Results section are inconsistent (e.g., intervention group denominators 549 vs 515 mentioned).

Based on these comments, I recommend a major revision. The authors are encouraged to reanalyze the data using models that account for clustering, include effect sizes with 95% confidence intervals, and run sensitivity analyses to address missing data or nonresponse bias. It would also strengthen the paper to adjust for multiple comparisons and to expand the discussion of limitations and interpretation. Once these revisions are made and the presentation of sample sizes and potential biases is clarified, the study has the potential to make a meaningful and valuable contribution to the literature.

Author Response

Comment 1: 

The program recruited clinicians by invitation and also provided $1000/month training reimbursement to participants. This high monetary incentive could influence responses. Need more reasoning for this bias in your limitations and if possible,  present baseline characteristics like age, gender, years in practice etc. to compare participants with non-participants and intervention with control clinicians. 

Response 1: 

We included demographic comparisons between cases and controls (those who filled out the surveys but did not receive the intervention). For the non-participants (those who did not fill out the surveys), we don't have age, gender and years in practice about them.  We have added a Table 3 that contains the demographic information for control and intervention groups for those individuals that filled out the survey at both time points. 

In the methods section, discussion of the training reimbursement was expanded to clarify the reasoning.  A clause was added to the limitations of the study in the last paragraph of the discussion, noting the potential bias and next steps given the training reimbursement.  

Comment 2: 

Another concern is the low and differential response rates i.e. baseline response rates differ widely, because intervention group is      54.1 percent whereas control group is 15.4 percent. The n=15 paired participants is very small for inferential claims. Also, some denominators in the Methods/Results section are inconsistent (e.g., intervention group denominators 549 vs 515 mentioned). 

Response 2: 

The denominators for the intervention groups are because the survey was administered tat 2 times points.  For the baseline survey the N= 549 and for the repeat survey the N=515 as multiple people had left their groups in the 1-year time period.  However, for tracking changes in emotional exhaustion and turnover intent, only individuals that filled out the survey at both time periods were included in the analysis. Please reference the paragraph immediately after Figure 1. We added a comment in the discussion about the difference in response rates between intervention and control groups with a likely explanation and added current data about well-being metrics in non-responders.  We also commented on the discrepancy in the limitations of the study. 

Comment 3: 

Based on these comments, I recommend a major revision. The authors are encouraged to reanalyze the data using models that account for clustering, include effect sizes with 95% confidence intervals, and run sensitivity analyses to address missing data or nonresponse bias. It would also strengthen the paper to adjust for multiple comparisons and to expand the discussion of limitations and interpretation. Once these revisions are made and the presentation of sample sizes and potential biases is clarified, the study has the potential to make a meaningful and valuable contribution to the literature. 

Response 3: 

Accounting for Clustering: In the final regression model, we accounted for clustering by adjusting for the state of the facility sites (MT, OR, and WA). This adjustment was made to ensure that any effects observed were not confounded by geographic clustering, thereby enhancing the robustness of our conclusions. 

Effect Size and Confidence Intervals: Since the analysis was performed using linear regression, we presented the effect size using mean estimates along with their standard error deviations in the linear regression analysis.  

Sensitivity Analysis for Missing and Non-response Data: For the regression analysis, we included only complete responses to maintain the integrity of the data. This means our conclusions are specifically applicable to the subset of participants who completed both pre- and post-intervention surveys. We acknowledge this limitation and have addressed it in the manuscript by emphasizing the need for careful interpretation given the exclusion of incomplete responses. This limitation is outlined in the discussion section to provide context for the findings and their applicability. 

Multiple Hypothesis Testing Adjustment: To address multiple hypothesis testing, we applied the Benjamini-Hochberg procedure, which adjusted the p-values obtained from multiple comparisons while maintaining a false discovery rate of 10%. In our tables, p-values that remain significant after adjustment are marked with an asterisk (*), clearly indicating statistical significance under these corrections. 

Round 2

Reviewer 1 Report

Comments and Suggestions for Authors

No additional comments.

Reviewer 3 Report

Comments and Suggestions for Authors

good for publication now